# A Review of Phase-Change Materials and Their Potential for Reconfigurable Intelligent Surfaces

**DOI:** 10.3390/mi14061259

**Published:** 2023-06-16

**Authors:** Randy Matos, Nezih Pala

**Affiliations:** Department of Electrical & Computer Engineering, Florida International University, Miami, FL 33174, USA; rmato006@fiu.edu

**Keywords:** phase-change materials (PCMs), metal-insulator transition (MIT) materials, reconfigurable intelligent surface (RIS), vanadium dioxide, GST, GSST

## Abstract

Phase-change materials (PCMs) and metal-insulator transition (MIT) materials have the unique feature of changing their material phase through external excitations such as conductive heating, optical stimulation, or the application of electric or magnetic fields, which, in turn, results in changes to their electrical and optical properties. This feature can find applications in many fields, particularly in reconfigurable electrical and optical structures. Among these applications, the reconfigurable intelligent surface (RIS) has emerged as a promising platform for both wireless RF applications as well as optical ones. This paper reviews the current, state-of-the-art PCMs within the context of RIS, their material properties, their performance metrics, some applications found in the literature, and how they can impact the future of RIS.

## 1. Introduction

In recent times, there has been increasing interest in phase-change materials (PCMs) as a promising avenue for developing adaptive materials that can revolutionize the field of metamaterial-based applications, including the reconfigurable intelligent surface (RIS). PCMs refer to a class of materials that can undergo a significant transformation in their material phase through external excitations such as conductive heating, optical stimulation, or the application of electric or magnetic fields, which in turn, results in changes in their electrical and optical properties. Notably, PCMs exhibit different degrees of volatility, with some being classified as volatile PCMs that require constant stimulation to maintain their transition state, while others are non-volatile PCMs that maintain their phase change even after the stimulus is removed until a certain level of stimulus is reached again. This variability in volatility is an important consideration when working with PCMs, and researchers are exploring new ways to leverage this unique property to create more dynamic and tunable materials. Examples of volatile PCMs that have been highly studied at the time of this review are vanadium oxides [1,2,3,4,5,6,7,8,9], titanium oxides [10,11,12,13], iron oxide (Fe_3_O_4_) [14,15,16], lanthanum cobaltite (LaCoO_3_) [17,18], niobium dioxide (NbO_2_) [19,20], and rare-earth nickelates (*R*NiO_3_) [21,22,23,24,25,26,27]. Chalcogenides (Ge_2_Sb_2_Te_5_/Ge_2_Sb_2_Se_4_Te, Sb_2_Se_3_, Sb_2_S_3_) [28,29,30,31,32,33] are among the most studied non-volatile PCMs in the recent literature. To cover any potential blind spots, we will cover PCM applications that fall under RF applications as well as the optical domain, particularly in the context of reconfigurable metasurfaces. As the literature suggests, using reconfigurable metasurfaces in wireless systems can improve system performance and reduce the probability of obstacles by increasing the number of controllable channels [34]. A recent survey also cited some early findings suggesting possible applications of RIS in the optical regime [35]. Applications of PCMs in other fields, such as passive thermal management of small spacecrafts [36], although very interesting, will not be covered in this review.

Figure 1 shows a graph of all the reviewed PCMs, along with their resistivity ranges and transition temperatures, found in the literature. Figure 1 plays an important role as it establishes a frame of reference for the potential applications of PCMs. Similarly, optical constants (refractive index *n* and extinction coefficient κ) of the reviewed PCMs are listed in Table 1. More importantly, the potential material limits of some PCMs are also laid bare.

One promising potential application for PCMs is RIS, which is expected to play an important role in the future of telecommunications as smart radio environments (SREs) are becoming essential to keeping up with the ever-increasing device density and data throughput [49]. While conventional methods to deal with these hurdles usually include moving to higher frequencies for larger bandwidths or to develop more complex multiplexing techniques for increased spectral efficiency, these methods are not without limitations. Higher frequencies mean that signals are more prone to attenuation. It has already been shown that as more and more systems enter mmWave territory for communications, humans and objects such as furniture can present blockages and penalize the link budget by up to 20–30 dB [50]. More complex systems also increase failure points or decrease efficiency. RIS can be implemented to overcome these setbacks, as it becomes possible to control the scattering and reflection of wireless signals and mitigate the negative effects of the propagation medium. Without the need for complex decoding, encoding, and RF frequency processing operation, RIS can manipulate the phase, amplitude, frequency, and even polarization of impinging signals to effectively control a wavefront [51,52,53,54,55,56,57]. SREs empowered by RIS devices can also reuse signals instead of creating new signals for data transmission, thus increasing efficiency [58,59]. Some utilization methods seen in the literature for RIS are software-defined metasurfaces, liquid crystal surfaces, and reconfigurable reflectarrays (RRAs) [51]. Methods of wavefront management and beam steering are still in their infancy, despite the research going on today. While mechanical beam steering has its uses [60], it is quickly left behind in SREs because it needs structures to be rotated or displaced and it is generally designed for specific polarizations. Electronic beam steering is the gold standard aspiration: manipulating waves with minimal power and no moving parts. The two major categories used to achieve electronic beam steering are lumped components and tunable materials. Lumped components refer to methods that use actual, physical devices on or in-between surface elements to modify surface currents by manipulating the resistance, impedance, and capacitance. The most common lumped components seen in the literature are RF MEMS, PiN diodes, and varactor diodes [61,62,63,64]. The literature also shows techniques using combinations of these lumped components [65]. Tunable materials, as the name suggests, use some form of excitation to change their structural properties, which in turn results in changes in their electro-magnetic properties. Liquid crystals, ferroelectrics, and graphene-based implementations have been proposed [66,67,68,69].

In the last few years, many researchers have reviewed different aspects of progress with PCMs. Zhang et al. discussed the role of crystallization in enabling neuro-inspired computing and universal memory using PCMs [70]. Wang et al. reviewed non-volatile photonic applications, such as phase-change photonic memory, phase-change photonic neuro-networks, phase-change metasurfaces, and phase-change color displays [71]. Shen et al. reviewed the state of fabrication and patterning of PCMs for future memory devices [72]. Cooley et al. reviewed the advancements in electrical contact technologies currently used when interfacing with PCMs to maximize performance and efficiency [73]. Ali et al. covered current cooling techniques with PCMs, nanofluids, and their combined use, leading to efficiency enhancements [74]. Zhang et al. reviewed advancements in thermal energy storage and heat transfer using PCMs [75]. Novel phenomena such as the thermo-capillary (Marangoni) effect was also studied in some PCM materials [76]. We can see that a comprehensive review for PCMs in the telecommunications and EM wavefront manipulation space is lacking. To complement the gap in existing reviews, this paper reviews the wide range of current, state-of-the-art PCMs, their material properties, their performance metrics, notable applications, and particularly how they can impact the future of reconfigurable metasurfaces in RF-THz-optical spectra. Section 2 of this paper will separate the PCMs into two major categories: volatile and non-volatile PCMs. Within these categories, we will take a deep dive into each relevant material and see how the phase transition takes place, briefly cover some of the work being done with said materials, and provide a brief discussion on such work. Section 3 will discuss the perspectives provided by this review, how they can inform the outlook of PCMs, and their impact on RIS and other metamaterial-based applications revealed in the course of the review.

## 2. Phase-Change Materials of Interest and Applications

### 2.1. Volatile PCMs

#### 2.1.1. Vanadium Dioxide

Vanadium dioxide (VO_2_) is the most studied of the many stoichiometric varieties of vanadium oxides due to its attractive phase transition characteristics [77]. VO_2_ is a phase-change material (also referred as metal-to-insulator transition material) that behaves as an insulator at room temperature but undergoes a phase transition to a metallic state when heated above ~69 °C. Below the threshold temperature, VO_2_ has a monoclinic crystal structure, which shifts to a rutile structure as the threshold temperature is approached and surpassed, as shown in Figure 2. To achieve this transition, various methods such as conductive heating, photo-thermal heating, Joule heating, and optical stimuli are used [78,79,80,81,82]. This transition has also been attained using static electric fields [83,84]. Due to the changes in resistivity and permittivity brought about by the transition, VO_2_ is an attractive material for applications in switching, optics, thermal diodes, antennas, waveguides, and resonators [79,82,85,86,87,88,89,90].

By taking advantage of this transition and based on their previous work [91,92], Matos et al. recently proposed a VO_2_-based platform that used a VO_2_ film on a heating matrix to create desired metallic patterns in real-time, very much like generating images on an LCD screen, and used it as a reconfigurable surface to modulate and steer incident EM waves within the 5G spectrum (Figure 3) [1,2,3,93]. The platform could reach VO_2_ MIT temperatures within 12 ms and revert to its insulating state within about one second. Unit-cell analysis showed that when the platform was used to develop a reflectarray antenna, it could achieve a reflection range of 310°. Full-wave array analysis of a 20 × 20 element reflectarray showed a directivity between 22–26 dBi with good beam scanning performance within ±50° elevation angle at 32 GHz. It was argued that the ultra-reconfigurable intelligent surface could provide a promising avenue for SREs and their development for 5G communications. The platform could effectively manipulate the phase, amplitude, frequency, and polarization to control the wavefront.

As an alternative to using VO_2_ as a main reflector on the entire device, more conventional “patch” approaches have also been explored. This approach was showcased by Shabanpour et al. with what they called a “VO_2_-based coding metasurface” (VBCM) (Figure 4a) [4]. The reported device had a three-patch pattern of VO_2_ that could be dynamically tuned to four reflection phases of 0, π/2, π, and π3/2, corresponding to four digital states of “00”, “01”, “10”, and “11”, respectively, by changing the bias voltage. The numerical analysis showed various reflection states and a maximum phase range of 260° between 0.3 and 0.6 terahertz (THz) for different combinations of active patches. Using the superposition theorem [94], they generated vortex-carrying orbital angular momentum (OAM). Further, by adding spiral phase distribution with gradient code sequences and using the convolution theorem [95], the device achieved scattering pattern shift functionality. The platform could meet the demands of future THz high data rate wireless communication systems and ultra-massive MIMO communication.

Song et al. presented three different structures based on the MIT properties of VO_2_ [5,6,7]. The transition properties found in VO_2_ also allow for manipulation of its absorption properties, which would prove attractive in applications for RF absorbers or transparent conductors. In [5] they presented a bifunctional THz device using patterned VO_2_ on an SiO_2_ spacer with gold strips and a VO_2_ film (Figure 4b). During the metal phase, the device behaved like an isotropic absorber within a frequency band of 0.52–1.2 THz and showed an absorptance of more than 90%. In the insulator phase, the same structure behaved as a linear polarization converter. Their numerical results showed that cross-polarized reflectance could reach 90% between 0.42 and 1.04 THz. This meant that the device could achieve both of its operation states within a similar frequency range. The authors attempted to convey that their architecture could provide another method to develop switchable photonic devices and functional components in phase-change materials. They reported a similar device in [6] but with transparent conductor properties (Figure 4c). The device comprised a gold ring on SiO_2_, a VO_2_ film, and a subwavelength gold mesh. When the VO_2_ film was in a conducting state, it had a strong absorption peak at 0.638 THz, which was caused by formation of localized magnetic resonances from opposite currents on the metallic ring and the VO_2_ film. When the VO_2_ film was in the insulating state, transmittance was 95.8% at 0.398 THz, making it transparent to transmission. This appeared to suggest that one could make an optically opaque material transparent by putting metallic rings on a subwavelength metallic mesh. The authors put forth that the behavior could be explained by the scattering cancellation scheme [96]. The results showed that the device could still maintain stability over a considerable range of incident angles, meaning that the authors were on track to show that their device could potentially support applications in terahertz energy farming, transparent conducting devices, modulating, and filtering.

The authors incorporated graphene into a similar device in [7]. This new device switched between a narrowband high-intensity absorber to a broadband absorber (Figure 4d). When VO_2_ was in the conducting state, their numerical analysis showed 100% absorption at 1.37 THz. While in the insulating state, broadband absorption >90% was shown for frequencies between 1.25 and 2.13 THz. Both TE and TM modes showed good absorption up until 60°. At incident angles greater than 60°, the device’s performance began to deteriorate. The inclusion of graphene made this device highly tunable since it had more than just “on” and “off” provided by the VO_2_ but also bias applied to the graphene. Their study showed the maximum absorption changing when the Fermi energy level was changed from 0.1 to 0.7 eV. For broadband absorption and an E_f_ of 0.1 eV, an absorption of 45.5% was obtained. At an Ef of 0.7 eV, the previously mentioned maximum for broadband absorption was obtained. The authors put forth that their device could be used for applications in intelligent absorbers, terahertz switches, and photovoltaic devices.

Integration of VO_2_ with Dirac semimetal films (DSFs) was also explored to create reconfigurable metamaterial devices. DSFs are 3D analogs of graphene with Dirac nodes, which are points in the Brillouin zone where the conduction and valence bands meet and the electrons behave as if they were massless, similar to photons. This results in several unique properties, such as high electron mobility. The Fermi energy can be adjusted by in situ electron doping. Wang et al. proposed a tunable, bifunctional, THz metamaterial device based on DSFs and VO_2_ [8]. The device consisted of five layers with a 0.5 μm thick VO_2_ film sandwiched between 10 μm thick polyimide layers, which were covered with 0.7 μm thick DSFs in the form of square split ring resonators rotated 90° with respect to each other (see Figure 5a). When in the dielectric phase, VO_2_ in the device allowed tunable, broadband, asymmetric transmission of linearly polarized THz waves. Conversely, when VO_2_ was in the metallic phase, the device served as a tunable, dual-directional absorber with perfect absorption in both forward and backward directions. For either case, tunability was achieved by varying the Fermi energy of the DSFs. The increased Fermi energy caused a blue shift in the spectra. Moreover, varying the conductivity of the VO_2_ layer in its metallic phase allowed the device to be switched to a reflector (with reflectance of 55%) or an absorber (with absorptance of 96.2%).

VO_2_ can also be simultaneously utilized for multiple purposes using a combination of material states. Using a very clever technique, Duan et al. presented a reconfigurable, quad-state optical system that was made possible by both the phase transition and hydrogenation of VO_2_ (Figure 5b) [9]. Their experimentally verified device showed a quadruple-state, dynamic plasmonic display that responded to temperature tuning, H-doping, and local phase transition control using electron doping. They then showcased an optical encryption scheme as an application using the proposed device. The device comprised Al/Al_2_O_3_ nano-disks that sat on a VO_2_ film, which was on a thick gold film. Pd nanoparticles were placed on the VO_2_ film to facilitate the hydrogenation and dehydrogenation of the VO_2_ film. The quad-states for the MIT of VO_2_ and the hydrogenated and dehydrogenated VO_2_ were described as m-VO_2_, r-VO_2_, H-VO_2_(M), and H-VO_2_(I). That is, the insulative state (monoclinic), the metallic state (rutile), lightly doped VO_2_, and heavily doped VO_2_. The hydrogenated VO_2_ was less dependent on temperature variations, which explained why there were only two hydrogenated states to their device. They achieved four color states using their device, which they believed was a product of resonance variation from the refractive index changes of VO_2_. The resonance changes seemed to be more pronounced for the m-VO_2_ and H-VO_2_(I) states, which the authors explained as being caused by m-VO_2_ and H-VO_2_(I) being insulators and functioning as dielectric spacers, along with the Al_2_O_3_. They saw this as a typical particle-on-mirror configuration [97]. They then demonstrated the versatility of their device by modulating the phase transition behavior using local electron doping. The authors revealed first-principles calculations and agreed that the mechanism responsible was that metals with higher Fermi levels could donate electrons across their interface with VO_2_, leading to increased carrier concentrations in VO_2_ [78,98]. Conversely, metals with lower Fermi levels had electrons donated from VO_2_, leading to increased transition temperatures. Finally, the authors showcased a novel, two-level, optical encryption methodology using their device. This device allowed the use of both temperature and hydrogenation to function as decryption keys. A QR code was created that appeared blank with no stimulation. When temperature was applied, the QR code pattern appeared. However, it was not complete unless the system was hydrogenated and the final “key” for the encryption appeared. The device was fully reversible, and the keys could be hidden by allowing the system to cool and loading the system with O_2_ to reverse the hydrogenation. Their platform, alongside further studies, may pave the way for new applications in optical information storage, optical encryption, and high-resolution optical and holographic display technologies, as stated by the authors.

This subsection discussed the various applications and properties of VO_2_ in the literature. VO_2_ is a phase-change material that undergoes a transition from an insulator to a metallic state when heated. It is used in applications such as switching, optics, thermal diodes, antennas, waveguides, and resonators. Some proposed applications are a VO_2_-based platform that creates metallic patterns in real-time to modulate and steer electromagnetic waves in the 5G spectrum and using VO_2_ patches to dynamically tune reflection phases for encoding information. VO_2_ can also be used to create devices with absorption and transparent conductor properties in the terahertz frequency range. Integration of VO_2_ with Dirac semimetal films allows for reconfigurable metamaterial devices. The presented applications demonstrate the versatility and potential of VO_2_ in various fields, including communications, photonics, and encryption technologies.

#### 2.1.2. Titanium Oxide

The titanium oxide (TiO_2_, Ti_2_O_3_, T_3_O_5_) series has been emerging as promising materials with applications as phase-change materials. They exhibit MITs as well as semi-conductor-metal transitions. The electrical conductivity of Ti_2_O_3_ increases from ~10^3^ to ~10^4^ S/m as it transitions from the insulator phase to metallic phase at ~450 K. However, unlike VO_2_, it does not undergo crystal structure transformation [99,100]. TiO_2_ undergoes metal-insulator transition caused by reduction under high-temperature and low oxygen partial pressure. Figure 6 shows the thermal dependence of the resistance of single-crystal TiO_2_ treated with different conditions (oxygen partial pressure, p_O2_, and temperature). It should be noted that the strong reduction (red curve) results in metallic conductivity ~10^11^ times higher than that of the bulk reference (blue curve) [101]. Details of the proposed mechanisms for this change are given in [101]. Ti_3_O_5_ undergoes first-order structural and magnetic-nonmagnetic transitions at 460 K on heating and 440 K on cooling, which results in 3 orders of magnitude change in its resistivity (Figure 6c) [102]. Some first-principles calculations exploring these transitions were investigated to help guide the growth and development of the series [103]. In these calculations, it was found that the monoclinic angle of β-Ti_3_O_5_ was 91.1°, it decreased to 90.3° during transition, and it ended at 91.2° in λ-Ti_3_O_5_. Naturally, as interest in the series grows, so do attempts at developing applications based on the titanate variants.

In [10], Ndjiongue et al. proposed a liquid crystal-based RIS cell with TiO_2_ nanodisks for beam steering in visible light communication (VLC). They first presented a general infrastructure needed to satisfy the needs of an RIS-based VLC beam steering device. They covered the influences of the refractive index, wavelength, and RIS physical depth by way of the single-slit Fraunhofer diffraction pattern [104]. Using that as a basis, they formulated the TiO_2_ and liquid crystal RIS. Both the liquid crystal medium and the TiO_2_ would provide different paradigms of phase-change capabilities for device control. Although no direct numerical or simulation-based analysis was directly presented for the device, some of the literature supported the claims and comparison tables were made that showed promise for the device. The authors argued that the device should be able to steer incident beams close to 90° while providing high tuning flexibility and dynamic control. They also claimed the device could operate between 2 to 5 V, compared to 1 to 3 kV for other meta lenses with artificial muscles in the literature.

The ultra-narrow bandgap of 0.1 eV for Ti_2_O_3_ and its photothermal conversion efficiency have sparked interest in photothermal applications. In [11], Cai et al. presented a two-step sputtering method for growing Ti_2_O_3_ films and analyzed their photothermal conversion performance and THz transmission properties across the MIT. They used simulated sunlight as a light source and an infrared camera to evaluate the photothermal effect on the Ti_2_O_3_ films being illuminated (Figure 7a). To measure the films’ response to THz transmission, THz-TDS was carried out while also heating the films for the MIT. The films exhibited a photothermal conversion efficiency of 90.45% and could achieve a THz tuning depth of nearly 45.8% with a wideband at 0.1–1 THz. Cai et al. concluded that their work could help realize solar light-tuning of THz waves and could open new avenues for THz applications.

When the metastability of λ-Ti_3_O_5_ was reported, an interest was sparked in discovering potential dopants to improve its stability. By performing first-principles calculations, a λ-Ti_3_O_5_-based electromagnetic wave absorber was presented in [12] by Fu et al. after finding a suitable dopant. The authors found that Li could be used a strong stabilizer, since λ-Ti_3_O_5_ was found to be metastable as a nanocrystal at room temperature. They presented absorption bandwidths of 7.9 and 7.4 GHz for λ-Ti_3_O_5_ and Li/λ-Ti_3_O_5_, respectively. Zhidik et al. developed a memristor using stoichiometric TiO_2_ and nonstoichiometric TiOx [13]. A structure of Mo-TiO_x_-TiO_2_-Cu was developed on a coaxial glass-metal kern. The TiO_2_/TiO_x_ films were grown using magnetron sputtering (Figure 7b). Hysteresis was detected once the applied bias was raised to 10 V and then decreased. A region of negative differential resistance appeared when the electrical forming of the obtained structures occurred in vacuum.

In summary, Ti_2_O_3_ undergoes a metal-insulator transition without a crystal structure transformation, and its electrical conductivity increases as it transitions from the insulator phase to the metallic phase. TiO_2_ undergoes a metal-insulator transition caused by reduction under high temperature and low oxygen partial pressure. Ti_3_O_5_ experiences first-order structural and magnetic-nonmagnetic transitions, resulting in a significant change in resistivity. The angles of the monoclinic crystal structure also vary during the transitions. Specific applications of the TiO_2_ and Ti_3_O_5_ variants, such as a liquid crystal-based beam steering device for visible light communications and the use of Ti_2_O_3_ films for photothermal applications, were also explored. Additionally, dopants are being explored to improve the stability of λ-Ti_3_O_5_, and a memristor was developed using TiO_2_ and TiO_x_ films.

#### 2.1.3. Iron Oxides

In its lower temperature phase, Fe_3_O_4_ is an insulator with a monoclinic structure connected to its charge order. It is a cubic ferrimagnetic oxide and has the highest magnetization, M = 4.2 μ_B_, below Curie temperature, T_C_ = 858 K. It exhibits a first-order structural phase Verwey transition (VT) at 124 K for stoichiometric films [105]. Half-metallic (−100% spin polarization) characterization of Fe_3_O_4_ with room temperature conductivity, σ = 200 (Ω·cm)^−1^, was predicted with band structure calculations [106,107]. This transition affects the properties of Fe_3_O_4_, such as resistivity, heat capacity, and magnetization. Figure 8 showcases a graphical representation of the transition of Fe_3_O_4_, its exchange interaction, and the density of states. This transition in Fe_3_O_4_ has naturally made it an attractive material for reconfiguring technologies due to its tunability via external stimulus.

In [14,16], Fe_3_O_4_ was used to improve microwave absorption. Yun et al. created a Fe_3_O_4_ microsphere-based film that functioned as a microwave absorber for varying thicknesses. The 2.6 mm thick composite film reached −54.38 dB for a frequency of 12.4 GHz. The maximum effective absorption bandwidth was extended to 4.1 GHz if the thickness was increased to 3 mm. In the thickness range of 2.5–5.0 mm, the effective absorption band covered the range of 4.2–13.8 GHz, thus occupying most of the test band. Based on the authors’ discussion, they put forth that the film consisting of many small Fe_3_O_4_ nanocrystals agglomerated to form a three-dimensional network, thus improving conductivity loss, and the surface nanocrystals generated by etching increased the contact area at the interface to enhance interface polarization. Etching of Fe_3_O_4_ microspheres produced defects which gave rise to additional dipole polarization, thereby increasing the dielectric loss capacity, and finally, the carbon shell enhanced interface polarization of the Fe_3_O_4_ microspheres. Conversely, Chen et al. opted to combine Fe_3_O_4_ with graphene to achieve the absorption properties. They reported the first three-dimensional, cross-linked Fe_3_O_4_/Graphene (3DFG). They reported ultra-broadband absorption from 3.4 GHz to 2.5 THz. Their device also showed durability, as it achieved stable absorptivity even after 200 multiple, repeated, compression/release cycles. Due to the VT previously discussed for Fe_3_O_4_, further tunability could be achieved by transitioning the Fe_3_O_4_.

Not long after, Guo et al. also combined Fe_3_O_4_ and graphene to achieve efficient electromagnetic shielding in [15]. Hybrid composite films of graphene nanosheets (GNSs)-Fe_3_O_4_/poly (vinylidene fluoride) were fabricated using a layer-by-layer process. The resultant composite films of GNSs (7.9 vol%)-Fe_3_O_4_ (5.4 vol%)/PVDF with a thickness of only 0.30 mm could exhibit excellent EMI shielding performance as high as 52.0 dB against the X-band and ultrahigh specific EMI SE of 173.3 dB mm^−1^. Their film also had the benefits of being lightweight, thin, and flexible, allowing for more versatile applications. The thermal effects of Fe_3_O_4_, could have been explored to probe how the VT would affect the film’s shielding capabilities.

In conclusion, in its lower temperature phase, Fe_3_O_4_ is an insulator with a monoclinic structure and it exhibits a VT at 124 K. It undergoes a first-order structural phase transition and has high magnetization below its Curie temperature of 858 K. Fe_3_O_4_ has been studied for its tunability and potential applications in reconfiguring technologies. The subsection also described two studies that used Fe_3_O_4_ for microwave absorption. Additionally, Fe_3_O_4_ was combined with graphene in another study to achieve efficient electromagnetic shielding. The tunability of Fe_3_O_4_, particularly through the Verwey transition, is highlighted as a potential avenue for further enhancing the properties and applications of Fe_3_O_4_-based materials.

#### 2.1.4. Lanthanum Cobaltite

LaCoO_3_ is a perovskite material that possesses an MIT for which there is increasing interest. LaCoO_3_ also has a spin-state transition around 100 K [109]. The MIT occurs around 500 K and a few things are observed as the temperature increases: CoO_3_ polyhedra move toward perfect octahedra, the rhombohedral distortion decreases (as evidenced by the increase in Co-O-Co angle toward 180° and the decrease in rhombohedral angle), and the Co-O distances increase following thermal expansion of the lattice [110,111]. Figure 9 depicts an illustration of the crystal lattice and the rotation undergone by the CoO_6_ polyhedra.

Currently, one of the main prospects of LaCoO_3_ seems to be in microwave absorption for possible applications in electromagnetic pollution, shielding, and stealth applications. In [17,18], LaCoO_3_ was studied by both doping and as a nanocomposite for improving microwave absorption properties. Yuan et al. prepared samples of LaCoO_3_ and Fe-doped LaCoO_3_ via a sol-gel method. The material properties were studied via XRD, EDS, and XPS while the electromagnetic properties were studied using a vector network analyzer (VNA). They learned that by changing the level of Fe doping, the complex permittivity could be reduced, the impedance matching could be improved, the reflection of electromagnetic waves could be reduced, and the wave absorption performance could be improved. Their analysis showed that LaCo_0.9_Fe_0.1_O_3_ had the most absorption performance, at a reflection loss of −38.99 dB and a bandwidth of 4.48 GHz. LaCo_0.95_Fe_0.05_O_3_ had the best absorption bandwidth, covering almost the whole X-band. Sun et al. also used a sol-gel method, creating a LaCoO_3_ nanocomposite with Bi_2_S_3_. The material was also studied using XRD and VNA. Material with 35 wt% Bi_2_S_3_ showed the best absorption, with a reflection loss of ~−17 dB using the waveguide method. While it was not directly explored in the literature, the absorption properties may be dynamically tuned using the MIT of LaCoO_3._

This subsection discussed the properties and potential applications of LaCoO_3_, a perovskite material that exhibits a metal-insulator transition (MIT) and a spin-state transition. As the temperature increases, the CoO_3_ polyhedra in LaCoO_3_ undergo changes, such as moving towards perfect octahedra, decreasing the rhombohedral distortion, and increasing Co-O distances. LaCoO_3_ is of increasing interest for microwave absorption applications, particularly in electromagnetic pollution, shielding, and stealth applications. The potential tuning of absorption properties using the MIT characteristics of LaCoO_3_ is suggested, although it was not directly explored in the literature.

#### 2.1.5. Niobium Dioxide

NbO_2_ is a material that undergoes a second-order Mott-Peierls transition and has recently gained more attention for applications in memristors, reflection, absorption, transistors, and sensors. NbO_2_ has a transition temperature of 1083 K [112], although studies have shown that NbO_2_ exhibits several activation energies from 165 to 870 K [46]. During the transition driven by a weakening of Nb dimerization without significant electron correlations, NbO_2_ goes from a body-centered tetragonal (BCT) structure to a rutile structure. Figure 10 shows an illustration of the different crystal orientations at low and high temperatures.

Due to the interest in NbO_2_ as a memristor, Novodvorsky et al. synthesized NbO_2_ for use in memristive thin films [19]. These films were grown using pulsed laser deposition (PLD) and were placed on a structure of Al_2_O_3_, Pt, and Nb_3_O_5_, topped with Nb islands. During their probing of the device, two voltage surges representing regions of negative differential resistance (NDR) were found at 1.25 and 1.75 V, both in the positive and negative region of the I-V curve. Although the device was measured at room temperature, the authors recognized that the I-V measurement effectively corresponded to two temperatures for NDR. That is to say, the device’s performance would be tunable with external thermal stimulus.

Reflective and absorptive surfaces have also seen some work recently where NbO_2_ is concerned. In [20], Shimabukuro et al. prepared a variety of suboxide thin films of Nb_x_Ti_1−x_O_2_, NbO_2_ and TiO_2−x_ using PLD. The layered stacks were designed and fabricated taking into account each of the film’s optical constants, which were obtained via ellipsometry. The device was designed as a six-layer stack with 2% reflectivity across the visible range (400–700 nm), even at moderate incidence angles. Furthermore, this six-layer coating displayed large and constant absorbance across the visible range, making it optically black. While the films were grown at different temperatures, the devices can be assumed to have been measured at room temperature. We know that the optical properties of NbO_2_ change with respect to oxygen content [113], and seeing how the lattice size changed during transition, it may be worthwhile to study to the effects of temperature on the film.

In summary, NbO_2_ is a material known for its second-order Mott-Peierls transition and it has potential applications in memristors, reflective surfaces, absorption, transistors, and sensors. NbO_2_ undergoes a transition from a BCT structure to a rutile structure with a transition temperature of 1083 K. The transition is driven by a weakening of Nb dimerization without significant electron correlations. The effects of temperature on the film’s optical properties and lattice size changes during the transition are suggested as potential areas for further study.

#### 2.1.6. Rare-Earth Nickelates

Rare-earth nickelates (*R*NiO_3_) are another area that has received increased interest due to their sharp MIT, changes in their crystal structure, and their electronic and magnetic properties [114,115,116,117]. The theoretical work on *R*NiO_3_ has been challenging and has only recently come to a consensus. Due to the sheer scope of the *R*NiO_3_ family, it is out of the purview of this paper to go over the history of the consensus, but it has been covered well in other places [118]. We will, however, generalize the latest consensus, but it cannot be extended to every material in the *R*NiO_3_ family. After extensive numerical analysis and DFT+U calculations, the common picture of the physical mechanism was found to be bond disproportionation. As the name suggests, this refers to a disproportion in the distance between similar bonds in the lattice, and as energy is added to the system, compressions and expansions of the lattice occur. This is described in Figure 11.

The MIT in *R*NiO_3_ (‘*R*’ being Pr, Nd, Sm, Eu, Gd, Tb, Dy, and Yb) has been scrutinized to customize the MIT properties using external electric fields [119,120], increased tensile and compressive strains [118,121,122], and rare-earth doping [123]. The most novel methodology is hydrogen doping, which was validated experimentally by Zhang et al. [124] using SmNiO_3_ (SNO) and a saltwater-mediated transition to hydrogenated SNO when bias was applied. This methodology was later explored by Yoo et al. [125] using first-principles calculations for the *R*NiO_3_ series, who found that the entire series should exhibit some form of MIT upon hydrogen doping. While research-based works are found aplenty with the *R*NiO_3_ series, experimental applications and even proposed applications are still in their infancy.

**Figure 11 micromachines-14-01259-f011:**
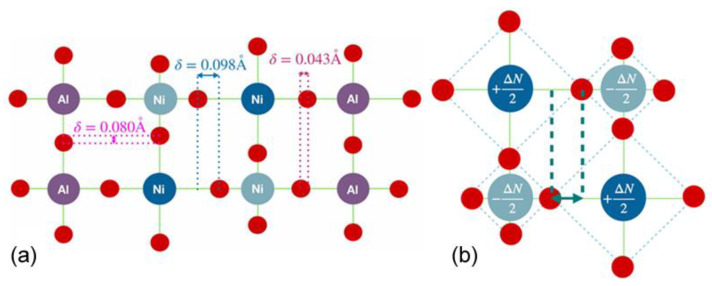
(**a**) Schematic showcasing bond disproportionation in NNO bilayer structures with NdAlO_3_ [126]. (**b**) Schematic of electronic and structural disproportionation in the insulating state in rare-earth nickelates: NiO_6_ octahedra electronically and structurally disproportionate in a checkerboard pattern [127].

In [21], Onozuka et al. investigated fluorine doping of NdNiO_3_ (NNO). Epitaxial thin films of NNO were grown on SrTiO_3_ substrates and the fluorination of the film resulted in electrical resistance changes greater than 6 orders of magnitude (Figure 12a). The optical absorption coefficient also showed a decrease as fluorination reaction times increased. This work could have been a precursor to [22], in which Sun et al. explored the thermochromic properties of NNO for applications in smart windows. NNO was grown using three different methods: sputtering, atomic layer deposition (ALD) and chemical solution deposition (CSD) (Figure 12b). The authors then studied the thermochromic properties of films in a basic solution of KOH using cyclic voltammetry (CV). Their results showed that the CSD-grown NNO showed the greatest degree of optical tunability due to the porosity of the film. The PVD- and ALD-grown films were denser than the chemically grown NNO and were subsequently less sensitive to the bleaching process. Sun et al. gave credence to the stability of the film showing promise for future applications in smart windows. They also detailed potential methods to improve the film’s properties, such as pore-generating agents or film patterning to grow more granular films and increase the surface area.

In [23], Liu et al. demonstrated a modified La_2_NiO_4_ (LNO) by doping Sr into LNO using a sol-gel method. The phase, microstructure electromagnetic properties, and microwave absorption were investigated. The LNO was doped at levels x = 0, 0.3, 0.5, and 0.75 for La_2-x_Sr_x_NiO_4±δ_. Using XRD, it was determined that the Sr-modified LNO adopted a tetragonal lattice, pertaining to the I4/mmm subgroup (Figure 12c). Their data showed that Sr doping caused an expansion of the unit cell, and the high intensity peaks shifted to lower angles as the concentration increased. Since Sr doping caused an increase in electrical conductivity and dielectric properties, the authors posited that the Sr-modified LNO showed promise for high-performance microwave absorption, which was numerically investigated via simulation. Their numerical analysis showed an optimal reflection loss of −21.07 dB with a broad range of 2.9 GHz for a material thickness of 1.9 mm. At the very least, this data strengthened their suggestion that the material could be explored for applications in microwave absorption.

In [24], Das et al. discovered pinned and bound modes in SNO. They grew epitaxial thin films of SNO on (LaAlO_3_)_0.3_(Sr_2_TaAlO_6_)_0.7_ (100) substrates using PLD. They then performed XRD analysis and terahertz time-domain spectroscopy (THz-TDS). The XRD analysis showed that as the film thickness increased, the unit cell volume decreased. The authors argued that this was due to an increase in oxygen vacancies, which led to a decrease in the concentration of Ni_3_+ cations, which are smaller than Ni_2_+ ions. The authors put forth that this implied an increase in oxygen stoichiometry as the thickness increased, leading to more stabilized films. THz-TDS was used to investigate the low-energy THz dynamics of the films between 10 and 300 K. The investigation showed a strong resonance peak at ~4 meV (peak A) and a broad peak at ~2 meV (Peak B). As the film thickness increased, peak A shifted to higher energy regimes and peak B became broader. At 110 nm, peak B was completely suppressed. This showed a very strong dependent behavior on both temperature and thickness. The authors argued that the two observed excitation modes were “pinned” and “bound” states of the charge-density-wave (CDW) condensate. They posited that the A mode appeared due to both impurities and commensurability of their samples, while mode B was caused by direct interaction of the impurities and the CDW condensate. Das et al. concluded that this effective method of controlling dual CDW excitation modes could be used in applications for CDW conduction, while also putting forth that the two resonance modes found in their study could be coupled with THz resonance modes of other artificially designed THz materials to control and guide light in different areas of metamaterials (Figure 12d).

To continue with investigations of SNO, in [25], epitaxial SNO films were grown in LaAlO_3_ substrates and investigated using a wide range of metrology that included scanning transmission electron microscopy (STEM), high-angle annular dark-field imaging (HAADF), annular bright-field imaging (ABF), X-ray energy dispersive spectroscopy (EDS), and electron energy-loss spectroscopy (EELS). The metrology revealed high-density 3D Ruddlesden-Popper (RP) faults, which the authors theorized could play a critical role in the MIT of SNO due to RP faults being a common crystal defect occurrence in nickelate and perovskite materials [128,129]. These RP faults formed two types of image contrast due to overlap in electron beam direction. The strong electronic-lattice distortions generated by RP defects also led to an MIT of 340 K. Zhong et al. presumed this study could guide the growth and performance control of nickelate perovskite oxide films, while also suggesting that it may be possible to deliberately harness RP faults to control physical properties in potential device applications.

This subsection discussed the RNiO_3_ series, their sharp MIT, changes in their crystal structure, and their potential applications. The consensus on the physical mechanism of MIT in RNiO_3_ is bond disproportionation, where the distances between similar bonds in the lattice undergo disproportionation with added energy. Hydrogen and rare-earth doping have been explored to customize the MIT properties. Fluorine doping of NdNiO_3_ films resulted in significant electrical resistance changes and a decreased optical absorption coefficient. The thermochromic properties of NdNiO_3_ were studied for smart windows, with chemically grown films showing greater optical tunability. Sr doping in La_2_NiO_4_ resulted in expanded unit cells and increased electrical conductivity and dielectric properties, suggesting potential for high-performance microwave absorption. Pinned and bound modes of CDW condensate were discovered in SmNiO_3_ films, offering potential applications in CDW conduction and metamaterials. High-density 3D RP faults were observed in SmNiO_3_ films, which could play a critical role in the metal-insulator transition and guide the growth and control of nickelate perovskite oxide films.

### 2.2. Non-Volatile PCMs

#### Chalcogenides

Chalcogenide materials, which are alloys based on the group 16 “chalcogen” elements, are compelling for active tuning capabilities due to their sharp changes in electronic and optical properties. This change comes about from their quick, reversible, and non-volatile switching between crystalline and amorphous states. This phase state also requires no additional energy to preserve the state. The phase change is described in Figure 13, where the crystal structure is shown to change depending on the energy input. It also illustrates the change in material resistance when set and reset. Accordingly, when a relatively longer and moderate pulse of energy is applied to heat the material just above the glass-transition temperature (T_g_), either electrically or optically, it assumes a crystalline structure. This is usually referred to as the “set” process. Conversely, when a short and high energy pulse is applied to heat the material above the melting temperature (T_m_), followed by abrupt cooling, it returns to its amorphous phase. This process is named the “reset.” Electrical (resistivity) and optical (refractive index) significantly changes between these two phases, which allows various reconfigurable/tunable electrical and optical devices. The required energy for set and reset transitions as well as the pulse durations and cooling rates vary for different materials. While this phenomenon was first discovered in 1968 [130], GeTe and TeGeSnAu were amongst the first materials to show fast recrystallization within a temperature range of 170–400 °C that was dependent on stoichiometry [131,132]. Soon after, other chalcogenide materials were studied, but Ge_2_Sb_2_Te_5_ (GST) and Ge_2_Sb_2_Se_4_Te (GSST) have seen the most attention. GST-based PCMs typically have crystallization temperatures of 100–150 °C and a melting temperature of ~600 °C with a very fast crystallization speed (<20 ns). The typical cooling rate is 1 °C/ns for the melt-quenching of most GST-based PCMs.

Due to GST and GSST being chalcogenides, they no longer need energy input to maintain their state once they crystallize; therefore, they are attractive materials for applications that will not have high switching frequency. In [28], Thompson et al. trained a neural network to simulate different geometrical parameters for 1D bars and 2D pillars of GST on alumina. Since neural network training is beyond the scope of this paper, we will focus only on the results. Their trained simulation looked for optimized geometry surfaces based on the 1D bars and 2D pillars. Computations were performed using rigorous coupled wave analysis (RCWA). The simulations found 80% to 95% reflectance and transmittance between the two phase states at λ = 2 µm for four permutations of light inclination angles (0°, 45°) and polarizations (s, p) (Figure 14a). For the 2D pillars, only one design showed 75% to 95% reflectance and transmittance for all azimuthal angles, φ, and for inclination angles θ ≤ 45. Thompson et al. posited that they had found, for the first time, an angle- and polarization-independent, high-contrast, switchable reflective/transmissive metasurface for short-wave infrared (SWIR). This presented a strong numerical foundation for future work to explore high-contrast SWIR devices based on GST.

In [29], Wang et al. demonstrated two device platforms using GST to realize phase-change antennas and metasurfaces. The first device used silver nanostrips as a nano-heater and to enable plasmonic antenna functions. The scattering spectra showed broad resonances above 700 nm and 30% modulation in observed resonance intensity. They observed backscatter efficiency during the 75% volume fraction that they believed was affected by the GST recrystallizing during quenching. Their results showed good agreement with the simulation. The following device was a metasurface that utilized destructive interference between two pathways. The 85 nm wide and 35 nm thick GST nanobeams were placed in an array with a period of 200 nm on top of a 60 nm thick silver strip. They first simulated the device using the finite-difference time-domain method, then verified the simulations by measuring the reflection spectra from the metasurface at normal incidence. At 700 nm, the device showed reflectances of 4.3% and 14.5% from the crystalline phase to the amorphous phase, respectively. At 755 nm, the reflectance change was a factor of 4.5. This demonstration showed that their device could serve as meta-atoms to construct large-scale dynamic metasurfaces in which each pixel can be individually accessed, thus opening up more opportunities to create a wide range of dynamic, random access metasurfaces that are capable of programmable and active wavefront manipulation.

In [30], Zhang et al. demonstrated a large-area reconfigurable metasurface using the phase-change capabilities of GSST, which was shown to have lower optical loss and a higher switching volume, to create thick PCMs with reversible switching capacity compared to GST [133,134]. The device consisted of GSST patterned as meta-atoms on a reflective metal heater. Electrical pulses of <12 V for 500 ms triggered crystallization, and short 20–23 V pulses for 5 µs triggered amorphization. Characterization of the metasurface showed absolute optical reflectance (ΔR) contrast of 40% at a 1.49 μm wavelength and relative reflectance modulation (ΔR/R) of up to 400% at a 1.43 μm wavelength. They also showcased quasi-continuous multi-state tuning by increasing the voltage pulses in increments of 0.1 V and measuring the reflectance spectra. Beam steering was also showcased, as the device functioned as a Huygens’ surface. In the crystalline state, the measured deflection efficiency in the 0th order was 24.8%. In the amorphous state, the measured deflection efficiency in the 1st order was 8.3%. It is likely that their device will provide a platform for assessing the crystallization kinetics of PCMs, which could represent a step forward in realizing fully integrated, chip-scale, reconfigurable optics. The authors suggested that their approach was transferrable to other emerging heater materials, such as silicon and graphene, for diversifying reconfigurable metasurface devices, which would be a boon to the field of PCMs.

In [31], Yue at al. demonstrated meta-atoms patterned from GSST on a CaF_2_ substrate to leverage third-harmonic generation (THG) in mid-wave infrared systems (MWIR) (Figure 14b). The device was characterized using a non-polarized, broadband light source with λ = 2–5 µm, which illuminated the metasurface and the transmittance was measured. The authors observed conversion efficiencies for the metasurface at λ_p_ = 4.36 µm up to 4.6 × 10^−7^, corresponding to a THG average power of 3.7 nW for an average pump power of 8 mW. The third-order nonlinear susceptibility (χ^(3)^) of amorphous and crystalline GSST films was also evaluated from the THG conversion efficiency. The effective values of χ^(3)^ were found to be (3.36 ± 0.41) × 10^−18^ m^2^V^−2^ in the crystalline state and (4.58 ± 0.59) × 10^−19^ m^2^V^−2^ in the amorphous state. Yue et al. found that this contrast made a promising material for new, dynamic, reconfigurable nonlinear optical architectures and presented a solution for the limitations of current MWIR systems.

In [32], Delaney et al. demonstrated a reconfigurable device for optical routing based on Sb_2_Se_3_ (SbSe). They deposited 23 nm of SbSe on top of a 220 nm silicon-on-insulator (SOI) rib waveguide, and used a 150 mW, 638 nm wavelength, single-mode diode laser to phase change the SbSe. They first characterized the system with a Mach-Zehnder interferometer (MZI). Spectra were collected every 25 pixels, and each pixel was spaced by 1000 nm along the MZI. The first demonstration showed optical tuning by phase changing the SbSe. A reversible 2π phase shift was obtained from 100 separate crystallization/amorphization pulses, and demonstrated that a 750 nm pixel size resulted in a resolution of 0.02π. The next demonstration was an optical router based on a multimode interference (MMI) patch (Figure 14c). The absolute MMI single-port transmissions before writing were −30.6 dB (top) and −30.8 dB (bottom) compared to −25.9 dB for a straight waveguide. After writing the full pixel pattern, a 92%:8% splitting ratio was found between the top and bottom outputs, respectively. There was a 5% difference between the simulated and experimental results, possibly due to the limitations in pattern registration onto the device, which the authors did not fully understand as of yet. Multiple pattern writings were performed with full resets in between to show full reconfigurability. The reset state showed no memory of the pattern, but the film would likely delaminate or develop void formations over many writings. As demonstrated by Delaney et al., their work could enable a range of complex photonic circuits needed for applications such as on-chip light detection, quantum information processing, artificial intelligence hardware, or optical tensor cores.

In [33], Liu et al. demonstrated a solid-state, programmable, color printing device that used the phase change in Sb_2_S_3_ (SbS) and the resulting refractive index change. An Al film was deposited on Si (001), then Si_3_N_4_/SbS/Si_3_N_4_ was deposited on the Al film. Crystallization was confirmed by measuring the reflectance and Raman spectra, while changing the laser power from 6 to 12 mW. The authors reported film damage and ablation at ≥20 mW. The n and k values were also explored using an ellipsometer. The entire sample was heated on a hotplate for 2 min at 300 °C, which crystallized the entire device. Then, a 780 nm femtosecond laser with a 100 fs pulse width was used to create 10 µm square color patches. The reflectance spectra were then simulated using the n and k values obtained, and they were in good agreement with the experimental values. The effect of SbS film thickness was also explored and showed a range of amorphous state colors. The authors then demonstrated writing and erasing of images to the device with a resolution of ~4.35 × 104 dots-per-inch (dpi), which corresponded to a pixel size of 500 nm × 500 nm (Figure 14d). The erasing took place by using a hotplate and heating the entire sample at once. There was a slight memory effect in the film because a color change took place when the image was erased, though no remnant of the image itself remained. This study presented compelling evidence of stepstones for the next generation of high-resolution color devices and optical encryption.

This subsection discussed the use of chalcogenide materials, specifically alloys based on group 16 elements, for their active tuning capabilities in electronic and optical properties. These materials undergo reversible and non-volatile switching between crystalline and amorphous states, requiring no additional energy to maintain their state. GST and GSST are chalcogenide materials that have received significant attention. GST-based PCMs exhibit fast crystallization speed and are suitable for various reconfigurable and tunable electrical and optical devices. Different device platforms and applications using chalcogenide materials were presented. Other chalcogenide materials were also discussed. SbSe has been used for optical routing devices, showcasing phase tuning and reconfigurability. SbS has been employed in a programmable color printing device, enabling high-resolution color images with erasable capabilities. Overall, chalcogenide materials offer unique properties for active tuning and reconfigurable devices in various domains such as optics, antennas, and metasurfaces. Their ability to undergo phase changes and exhibit different optical and electrical properties makes them valuable for future applications in advanced technologies.

## 3. Perspectives and Outlook

From this review, we can see that PCMs offer unique solutions for next-generation RF and optoelectronic devices and technologies. Their use as bulk materials, thin-film materials, and in various RF and optoelectronic devices suggests that they provide a sizable contribution and importance to modern communication devices. Traditionally, PCMs have relied mainly on electrical, thermal, and/or optical pulses to achieve the changes in electrical and optical properties that are a product of the transition. The rapid transition together with strong property contrast between different solid states have inspired PCM-based metasurfaces that rely on switchable electronic and optical properties. These metasurfaces triggered the promising applications described in this review, such as reflectarray unit cells, antennas, switchable perfect absorbers, transparent conductors, beam steering, and color displays. All the PCM-based devices discussed in this work are tabulated in Table 2. Despite our findings in this review, there are still potential research directions that can be explored to expand the applications and use cases of the PCMs discussed herein. From the various temperature ranges alone, we can see that some materials would fare far better in low-temperature applications and some in high-temperature applications. The transition is also usually accompanied by a change in magnetic properties, which can potentially lead to novel applications that are beyond the scope of this review. Analysis of the switching volume as a function of oxide thickness could enlighten the areas for which some PCMs would be better suited. We saw thermal damage on some PCMs due to heat cycling, for which melting-induced healing can be explored. Novel growth and fabrication techniques can be explored in order to minimize PCM damage or to improve stoichiometry and roughness during reactive and wet etching. Doping of different ions and how they affect the transition is still a largely unexplored area in most PCMs, as even some of the transition mechanisms are still being deliberated. As PCM-based metasurfaces continue to be developed, the trend towards multifunctionalities that are compact and highly integrated should continue. Considering the easy-to-integrate properties of PCMs, when these hurdles are overcome, PCMs could become the next ubiquitous technology to usher in widespread SREs and optical applications.

## 4. Conclusions

In conclusion, this paper has reviewed the state of PCMs, their impact on reconfigurable metasurfaces, and various applications found in the literature. Various PCM-based materials, such as VO_2_, chalcogenides (Ge_2_Sb_2_Te_5_, Ge_2_Sb_2_Se_4_Te, Sb_2_Se_3_, Sb_2_S_3_), titanium oxides, and rare-earth nickelates (*R*NiO_3_) were discussed. This work covered the transition mechanisms, properties, and some applications currently found in the literature for each of the aforementioned PCMs. From that, it was gleaned that PCM-based devices have been proposed and developed in areas such as reflectarray unit cells, antennas, switchable perfect absorbers, transparent conductors, beam-steering, and color displays. Evidently, we see that there is room to improve PCMs via novel techniques in growth and fabrication, as material quality plays a vital role in PCM properties. Integration with established and novel fabrication techniques also requires further work for widespread adoption of PCM devices. With continued effort, researchers may yet find ways to overcome the hurdles discussed in this review, which will illuminate the path forward to continue improving our current and future communication systems.

## Figures and Tables

**Figure 1 micromachines-14-01259-f001:**
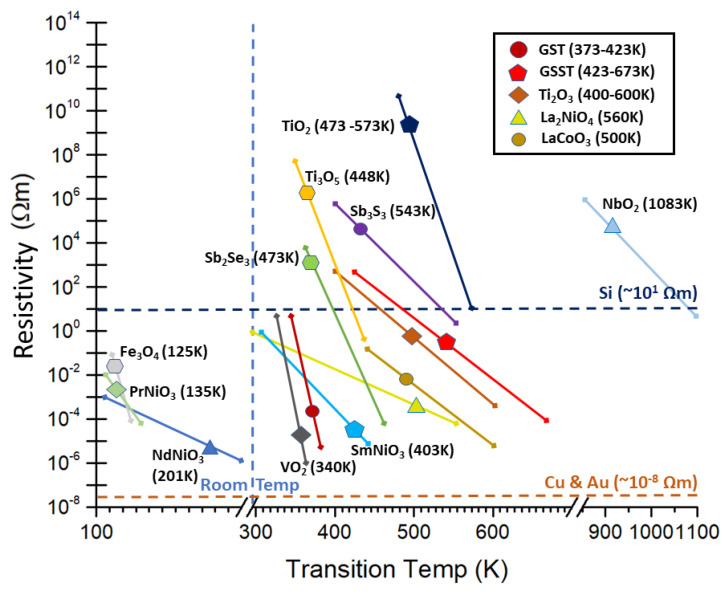
Resistivity ranges and transition temperatures for all reviewed PCMs in this work. Silicon, gold, and copper are also shown as guidelines.

**Figure 2 micromachines-14-01259-f002:**
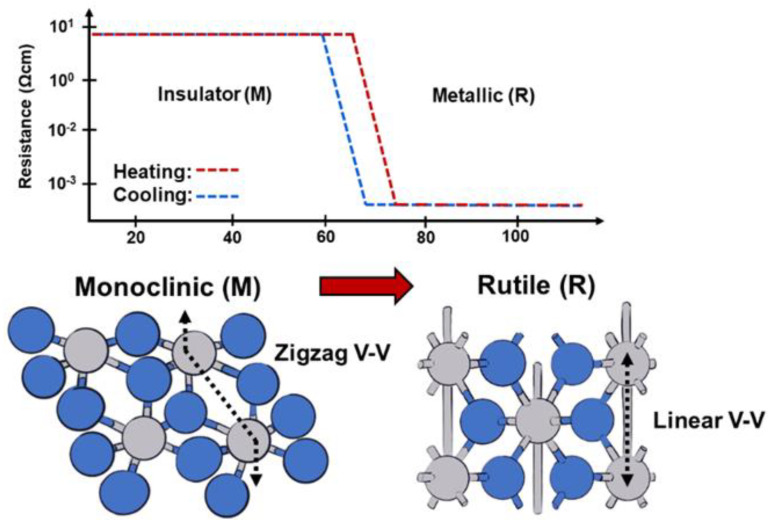
Representation of the crystal structure of VO_2_ and how it changes from a monoclinic to rutile structure during MIT. The associated resistance drop and hysteresis can also be seen above.

**Figure 3 micromachines-14-01259-f003:**
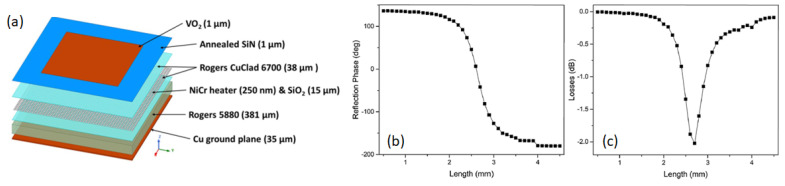
(**a**) Reflectarray unit cell using VO_2_ and an addressable heating matrix. (**b**) Reflection phase curve and (**c**) loss curve of the simulated structure for 32 GHz [1].

**Figure 4 micromachines-14-01259-f004:**
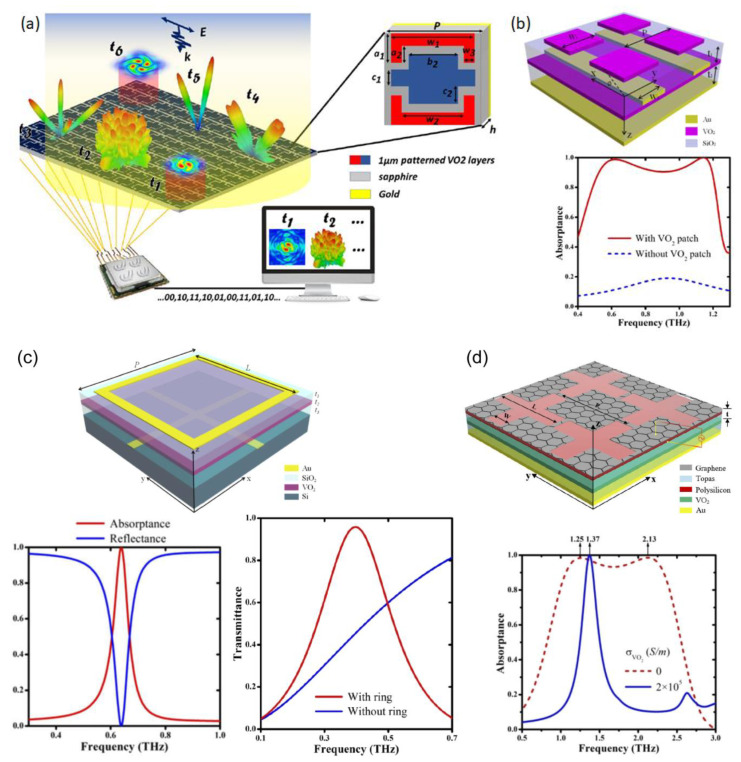
(**a**) Sketch representation of reprogrammable VO_2_-based coding metasurface controlled by an FPGA platform. Different spatial coding patterns can be encoded onto the structure simultaneously through a computer-programmed biasing network at distinct moments of t_1_, t_2_, t_3_… [4]. (**b**) (Top) Schematic of the switchable metasurface, consisting of periodic square-shaped VO_2_, SiO_2_ spacer, gold strip, VO_2_ film, SiO_2_ spacer, and the bottom gold film. (Bottom) Calculated absorptance of the designed absorber with (red solid line) and without (blue dashed line) the VO_2_ patch when the thicknesses of VO_2_ and SiO_2_ are 0.08 and 41 μm, respectively [5]. (**c**) (Top) Three-dimensional schematic of the switchable metamaterial consisting of square gold ring, gold cross, SiO_2_ spacer, and VO_2_ film on Si substrate; (bottom left) the simulated results of absorptance when VO_2_ is in the conducting state; (bottom right) the simulated transmittance with (red line) and without (blue line) the metallic ring when VO_2_ is in the insulating state [6]. (**d**) (Top) Three-dimensional diagram of the bifunctional metamaterial absorber based on graphene and VO_2_, using polysilicon, a Topas spacer, and VO_2_ and gold films; (bottom) the absorptance spectra of different VO_2_ state. Blue solid line (red dash line) denotes the result when VO_2_ is in the conducting (insulating) state [7].

**Figure 5 micromachines-14-01259-f005:**
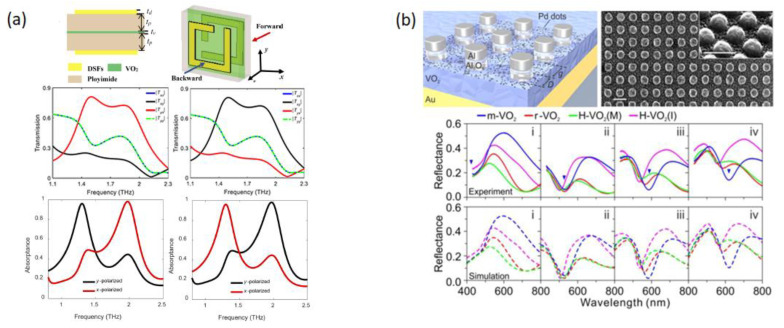
(**a**) (Top) Schematics of the proposed DSF-VO_2_ bifunctional terahertz metamaterial; (middle panel) transmission coefficients of linearly polarized waves in (left) backward (−z) and (right) forward (+z) directions; (lower panel) the calculated absorptance spectra for x- and y-polarized waves incident along the (left) backward and (right) forward directions [8]. (**b**) (Top left) Schematic of the quadruple-state dynamic color display. Al_2_O_3_/Al nanodisks with different diameter (D) and interparticle gap (g) reside on a VO_2_/Au mirror substrate; (top right) overview SEM image of a palette square. Inset: enlarged tilted view of the SEM image. Scale bar: 200 nm. (Bottom) Experimental and simulated reflectance spectra of the selected color squares at the four different states [9].

**Figure 6 micromachines-14-01259-f006:**
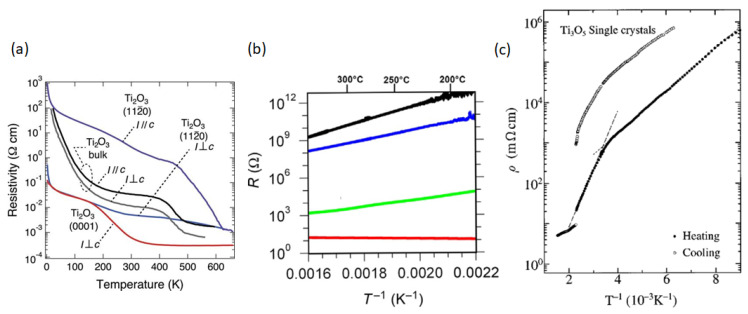
(**a**) Temperature dependence of resistivity for Ti_2_O_3_ films. The blue and purple lines indicate the a-axis in Ti_2_O_3_ films measured along I ⊥ c and I//c directions, respectively. The red line indicates the c-axis in Ti_2_O_3_ films. The bulk Ti_2_O_3_ data are also shown for comparison. The black and gray lines indicate the measurement directions along I//c and I ⊥ c directions, respectively [100]. (**b**) Thermal dependence of resistance of TiO_2_ single crystals treated with different conditions. The blue curve is the untreated reference, the black curve is oxidized TiO_2_ (p_O2_ = 200 mbar, 200 °C), the green curve is slightly reduced TiO_2_ (p_O2_ = 10^−9^ mbar, 500 °C), and the red curve is strongly reduced TiO_2_ (p_O2_ = 10^−12^ mbar, 1000 °C) exhibiting metallic conductivity. [101]. (**c**) Temperature dependence of the electrical resistivity of Ti_3_O_5_ [102].

**Figure 7 micromachines-14-01259-f007:**
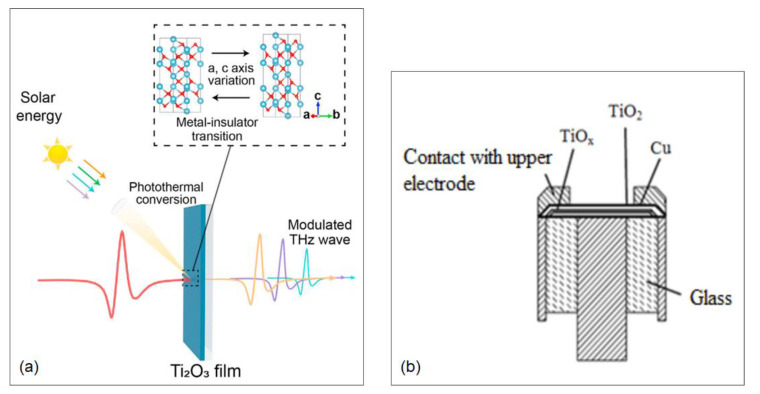
Devices and metasurfaces using TiO-based materials. (**a**) Schematics of a metasurface based on Ti_2_O_3_ being tuned under sunlight illumination to modulate THz waves [11]. (**b**) Schematic of memristor structures on kern-type substrate using a multilayer TiO_2_ and TiO_x_ metafilm [13].

**Figure 8 micromachines-14-01259-f008:**
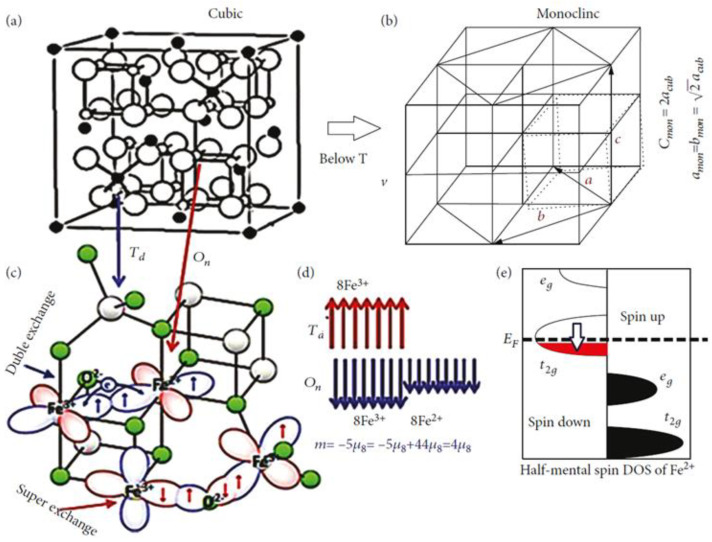
Cubic inverse spinel structure of Fe_3_O_4_ with tetrahedral and octahedral sites above T_V_ (**a**) and distorted monoclinic crystal structure below T_V_ (**b**). The double and super-exchange interaction mechanisms (**c**). Sketch of the localized structure of the spins at tetrahedral and octahedral sites (**d**). The density of states occupied by electrons of Fe^2+^ ions at octahedral sites (**e**) [108].

**Figure 9 micromachines-14-01259-f009:**
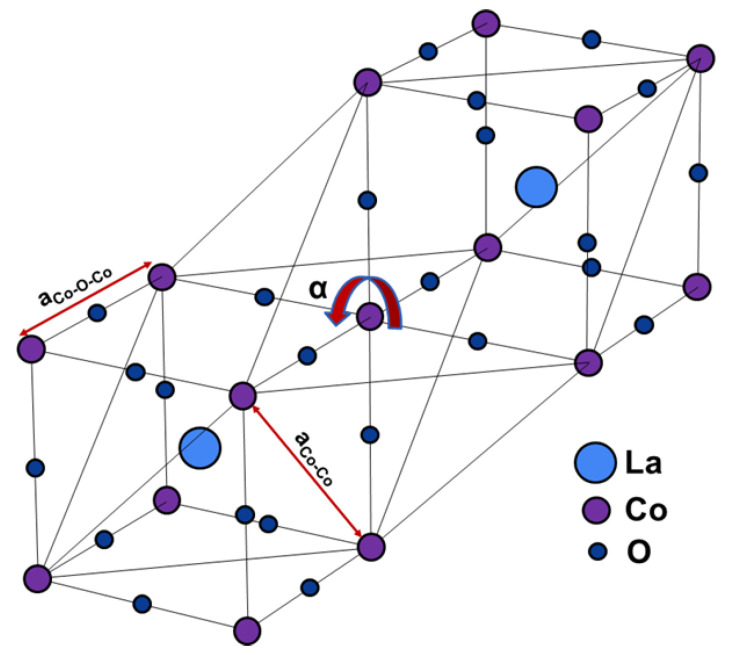
Structure of LaCoO_3_ before transition. Only one CoO_6_ polyhedron is shown for clarity. As the temperature increases, the CoO_6_ rotates and the distances between Co-Co and Co-O-Co shorten.

**Figure 10 micromachines-14-01259-f010:**
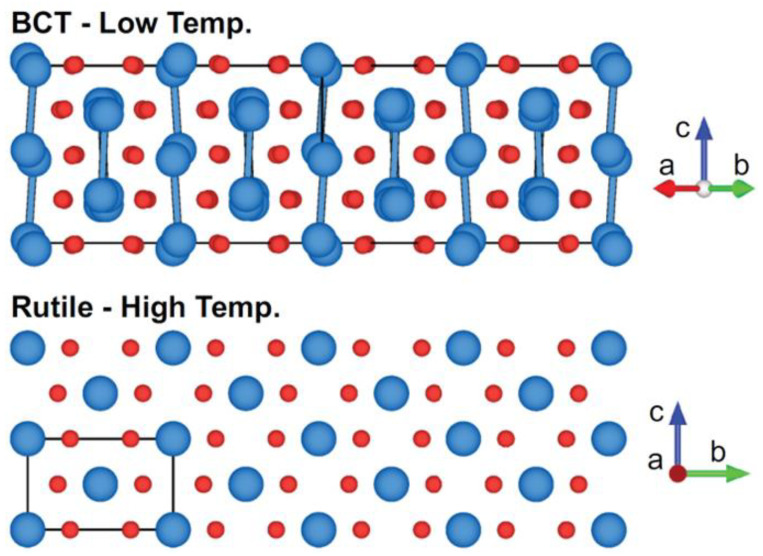
Low-temperature BCT and high-temperature rutile crystal structures of NbO_2_ [112]. Reprinted figure with permission from [112], Copyright 2019 by the American Physical Society.

**Figure 12 micromachines-14-01259-f012:**
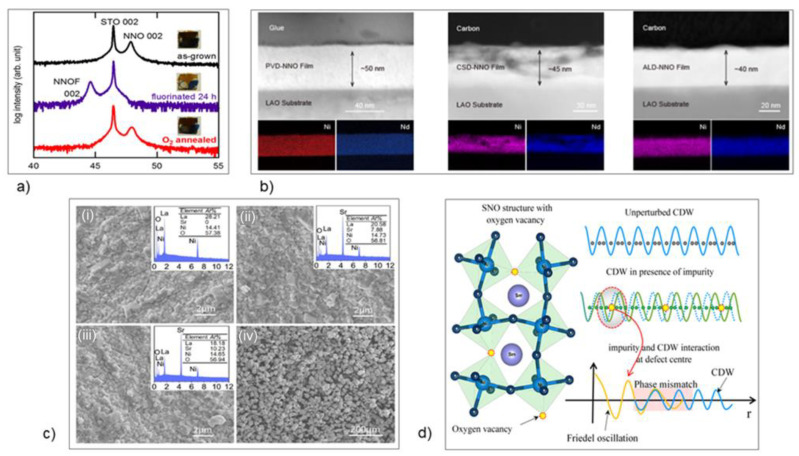
Devices and metasurfaces using rare-earth nickelates (RNiO_3_). (**a**) XRD 2θ−θ patterns of as-grown, 24 h fluorinated, and oxygen-annealed (450 °C for 5 h) NdNiO_3_ films [21]. (**b**) Cross-sectional HAADF-STEM and the corresponding EDX mappings of Ni and Nd elements. The CSD-NNO/LAO thin film is porous, while the other two are dense [22]. (**c**) SEM micrographs of the fracture surface of La_2-x_Sr_x_NiO_4_ ceramics [23]. (**d**) Schematic representation of SNO structure with random oxygen vacancies, which act as impurity centers. The unperturbed CDW mode gets entangled with this impurity-mediated Friedel oscillation (FO), centered at the impurity, and the phase mismatch between FO and CDW can be witnessed by the newly modulated lattice arrangement [24]. (**a**) Reprinted with permission from [21]. Copyright 2017 American Chemical Society. (**b**) Reprinted with permission from [22]. Copyright 2021 American Chemical Society. (**d**) Reprinted with permission from [24], Copyright 2020 by the American Physical Society.

**Figure 13 micromachines-14-01259-f013:**
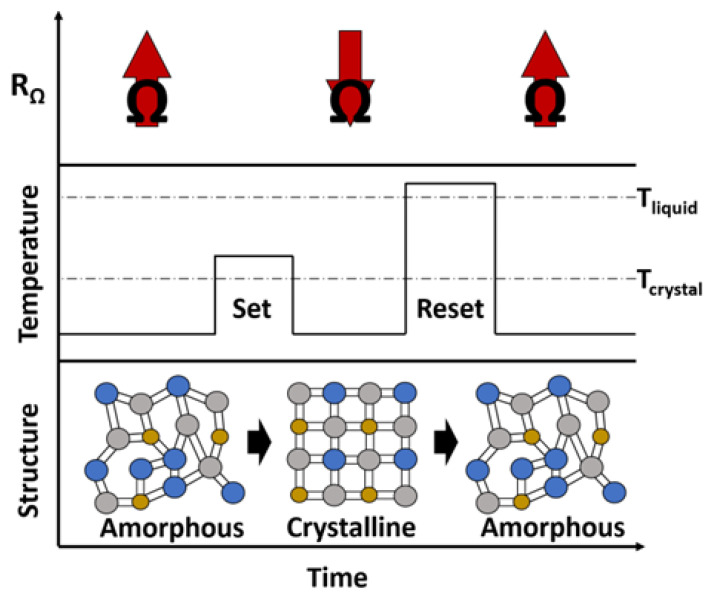
Representation of the phase change that occurs in chalcogenide materials. Thermal pulses either crystallize or amorphize the material to ‘set’ or ‘reset’ its high and low resistance phases.

**Figure 14 micromachines-14-01259-f014:**
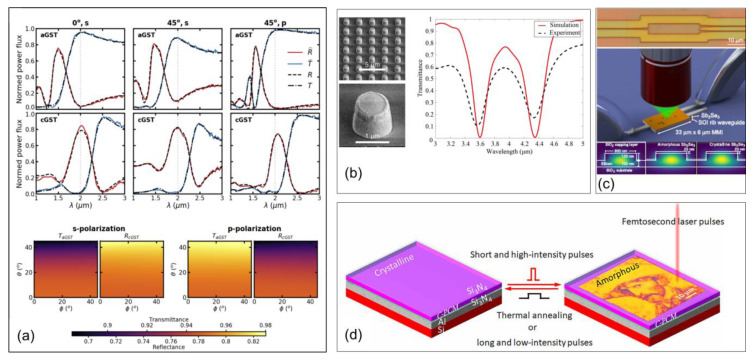
Devices and metasurfaces using chalcogenide materials. (**a**) Spectral plots of the 2D pillar array metasurface optimized as an angle- and polarization-independent switchable mirror using Φ data [28]. (**b**) SEM scan of the fabricated crystalline-GSST metasurface, with an enlarged image of an individual meta-atom. Simulated and measured (FTIR) transmittance of the GSST metasurface [31]. (**c**) Silicon photonic MMI. Scale bar: 10 µm. Illustration of MMI with thin Sb_2_Se_3_ PCM patch and optical writing of pixel patterns onto the device using a microscope. Illustration of the geometry of 500 nm wide silicon rib waveguides without PCM patch and with Sb_2_Se_3_ patch in amorphous and crystalline states [32]. (**d**) Schematic of the rewritable device consisting of antimony trisulfide PCM switched between crystalline (C-PCM) and amorphous (A-PCM) states. The thin film consists of Si_3_N_4_ (5 nm)/Sb_2_S_3_ (t nm)/Si_3_N_4_ (5 nm)/Al (100 nm) [33].

**Table 1 micromachines-14-01259-t001:** Refractive index *n* and extinction coefficient κ of the reviewed PCMs.

PCM	Optical Constants (n, κ)				Ref.
	Insulating/Amorphous	Measured Wavelength λ (nm)	Metallic/Crystalline	Measured Wavelength λ (nm)	
VO_2_	3.1, 0.4	500	2.8, 0.7	500	[37]
GST	3.9, 1.5	700	1.6, 3.7	700	[38]
GSST	3.5, 1.8 × 10^−4^	1000	5.1, 1.18	1000	[39]
SbSe	3.47, 0.71	500	3.9, 4.3	500	[40]
SbS	3.15, 0.13	621	3.55, 0.8	621	[41]
TiO_2_	2, 0.02	700	2.6, 0.15	700	[42]
Ti_2_O_3_	1.2, 0.6	621	N/A	N/A	[43]
Ti_3_O_5_	2.8, 1.9	621	0.1, 1	621	[44]
LaCoO_3_	2.6, 0.65	700	2.2, 0.8	700	[45]
NbO_2_	2.35, 2.57	621	N/A	621	[46]
La_2_NiO_4_	1.35, 1.5	621	1.95, 1.1	621	[47]
SmNiO_3_	3.2, 2.4	2000	2.4, 2.5	2000	[48]

**Table 2 micromachines-14-01259-t002:** PCMs explored in this work alongside the devices, their properties, and applications explored or suggested in their respective work.

PCM	Device Type	Properties	Applications	Refs.
VO_2_	VO_2_-based unit cell	- UC reflection range of 310°. - 20 × 20 full-wave array showed directivity between 22 and 26 dBi with beam scanning within ±50° elevation angle at 32 GHz	Reflectarray antenna	[1,2,3]
VO_2_	VO_2_-based coding metasurface	- Reflection phase range of 260° between 0.3 and 0.6 THz	THz communications systems and MIMO systems	[4]
VO_2_	Bifunctional VO_2_-based THz device	- Metal phase: isotropic absorber, absorptance of more than 90% between 0.52 and 1.2 THz. - Insulator phase: linear polarization converter, cross-polarized reflectance can reach 90% between 0.42 and 1.04 THz.	Switchable photonic devices	[5]
VO_2_	Simultaneous absorber and transparent conductor	- Metal state: absorber, absorption peak at 0.638 THz. - Insulator phase: transparent conductor, transmittance is 95.8% at 0.398 THz	Terahertz energy farming, transparent conducting devices, modulating, and filtering	[6]
VO_2_	Terahertz bifunctional absorber	- Metal phase: 100% absorption at 1.37 THz - Insulator phase, absorption >90% between 1.25 and 2.13 THz	Intelligent absorbers, terahertz switches, and photovoltaic devices	[7]
VO_2_	Broadband switchable terahertz HWP/QWP	- Bandwidth of 0.8–1.2 THz - HWP, PCR and DoLP >0.96 and >0.94, respectively. - QWP: phase difference ≈ −90° ellipticity close to −1.	Switchable, reconfigurable metasurfaces	[8]
VO_2_	Reconfigurable multistate optical system	- Four different color states based on changes in refractive index in metal phase, insulator phase, hydrogenated state, and dehydrogenated state.	Optical information storage, optical encryption, and high-resolution optical and holographic displays	[9]
GST	Switchable metasurface reflector	- 1D bars: 80% to 90% reflection and transmittance between two phase states - 2D pillars: 75% to 95% reflection and transmittance between two phase states	Switchable reflective/transmissive metasurface for SWIR	[28]
GST	GST-based antenna and metasurface	- Plasmonic antenna: resonance >700 nm, 30% modulation in resonance intensity - Metasurface: reflectance of 4.3% and 14.5% from crystalline to amorphous phase @ 700 nm, respectively. At 755 nm, reflectance change is factor of 4.5.	Meta-atoms for large scale metasurfaces, programmable active wavefront manipulation	[29]
GSST	Electrically reconfigurable non-volatile metasurface	- ΔR contrast of 40% at λ = 1.49 μm - ΔR/R up to 400% at a λ = 1.43 μm - Quasi-continuous multi-state - Beam steering	Chip-scale reconfigurable optics	[30]
GSST	Nonlinear mid-infrared metasurface	- χ(3) = (3.36 ± 0.41) × 10^−18^ m^2^V^−2^ for crystalline state and (4.58 ± 0.59) × 10^−19^ m^2^V^−2^) for amorphous state	Reconfigurable nonlinear optical architectures, MWIR systems	[31]
SbSe	Nonvolatile programmable silicon photonics using Sb_2_Se_3_	- Reversible 2π phase shift - Reconfigurable	On-chip light detection, photonic quantum technology, artificial intelligence hardware, optical tensor cores	[32]
SbS	Rewritable color nanoprints in antimony trisulfide films	- 40,000 dpi programmable color display - Color variety - Rewritable	High-resolution colordisplay devices, optical encryption	[33]
TiO_2_	Liquid crystal-based RIS cell with TiO_2_ nanodisks	- 2 to 5 V for ≈90° beam steering	VLC beam steering	[10]
Ti_2_O_3_	Photothermal conversion of Ti_2_O_3_ film for tuning terahertz waves	- Photothermal conversion efficiency of 90.45% - THz tuning depth of 45.8% - Wideband at 0.1–1 THz	Solar light-tuning of THz waves	[11]
Ti_3_O_5_	λ-Ti_3_O_5_-based electromagnetic wave absorber	- Absorption bandwidth of 7.9 and 7.4 GHz for λ-Ti_3_O_5_ and Li/λ-Ti_3_O_5_, respectively	Electromagnetic absorber	[12]
TiO_2_/TiO_x_	Memristor using TiO_2_/TiO_x_ thin films	- Hysteresis at 10 V - Negative differential resistance	Memristors	[13]
NNO	Fluorinated perovskite nickelate	- Fluorinated films become highly insulating with 2.1 eV bandgap	Thermal or atmospheric sensors	[21]
NNO	Thermochromic NNO films	- Thermochromic tunability by CSD	Smart windows	[22]
LNO	Sr-modified LNO	Tetragonal lattice - XRD high-intensity peaks at lower angles - Higher conductivity than LNO - For 2.9 GHz, reflection loss of −21.07 dB	Microwave absorber	[23]
SNO	Pinned and bound modes in SNO	- Strong dependent behavior on temperature and film thickness - Pinned and bound states of CDW condensate	CDW conduction, optical waveguides	[24]
SNO	SNO films with RP faults	- Two types of image contrast - MIT at 340 K	Basis for growth performance for nickelate oxide films, suggests possibility of harnessing RP faults to control physical properties	[25]

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
