# Peer review of "A Review of Phase-Change Materials and Their Potential for Reconfigurable Intelligent Surfaces"

_micromachines, 2023, doi:10.3390/mi14061259_

Round 1
Reviewer 1 Report
The revised manuscript provides a comprenhensive review of PCMs used as RIS. The paper is well written and structured and will be of interest for readers working on this topic. I congratulate the authors for their work; I have enjoyed reading through these lines.
From my point of view, only minor English editing is required to check for typos. After this revision is done, the paper can be accepted for publication.
An additional comment, I am curious about potential applications of RIS for space exploration. Currently, there is wide interest on the use of PCMs to passively control the temperature of small spacecraft, and different streagies are envisaged to improve their performance, in particular, the use of the thermocapillary effect. Do RIS systems require faster phase change in some applications so that they can benefit from thermocapillary flows in some applications? If the authors consider this disucssion of interest, it can be shortly included in the Introduction.
Minor revision checking for typos.
Author Response
Response to the Reviewers’ Comments
Journal: Micromachines
Manuscript ID: micromachines-2424004
Title: A Review of Phase Change Materials and their Potential for Reconfigurable Intelligent Surfaces
Authors: Matos, Randy; Pala, Nezih
We sincerely thank the reviewers for their invaluable feedback. We have revised our manuscript to address the reviewers’ comments. Below are our detailed responses to the comments. The parts added to the revised manuscript are highlighted in yellow.
Reviewer #1:
The revised manuscript provides a comprehensive review of PCMs used as RIS. The paper is well written and structured and will be of interest for readers working on this topic. I congratulate the authors for their work; I have enjoyed reading through these lines.
From my point of view, only minor English editing is required to check for typos. After this revision is done, the paper can be accepted for publication.
An additional comment, I am curious about potential applications of RIS for space exploration. Currently, there is wide interest on the use of PCMs to passively control the temperature of small spacecraft, and different strategies are envisaged to improve their performance, in particular, the use of the thermocapillary effect. Do RIS systems require faster phase change in some applications so that they can benefit from thermocapillary flows in some applications? If the authors consider this discussion of interest, it can be shortly included in the Introduction.
Response: Thank you very much for the kind words. The paper was reviewed for English edits and typos. To address your secondary comment, applications other than reconfigurable surfaces, although very interesting, are not within the scope of this paper. To further clarify the scope, we added the following sentence in line 44:
Applications of PCMs in other fields, such as passive thermal management of small spacecrafts [36], although very interesting, will not be covered in this review.
Yet, to inform the readers about thermocapillary (Marangoni) effect in PCMs, we added the following sentence in line 99 with the reference to a recent work:
Novel phenomena such as thermo-capillary (Marangoni) effect was also studied in some PCM materials [76].

Reviewer 2 Report
Reconfigurable intelligent surface has emerged as a promising platform for both wireless RF applications as well as optical ones. This paper reviews the current state of the art of PCMs within the context of RIS, their material properties, their performance metrics, some applications found in literature, and how they can impact the future of RIS. Major revision is necessary according to the following comments.
1. The author should compare the basic properties of several main PCMs mentioned in reconfigurable intelligent surfaces, in the form of figure or table.
2. In introduction section, the author should point out the differences between this review and similar topic reviews, highlighting the importance and timeliness of this review.
3. The last paragraph of the introduction cannot simply describe the basic structure of this review. Major revision is very necessary referring to other reviews.
4. A summary should be added at the end of each subsection.
5. Pay attention to the logical coherence between chapters and paragraphs.
6. In section 2.2.1 Chalcogenides, the author is more like a detailed list of five literatures [29-33], and should simplify the basic results and express more of his own opinions.
Minor editing of English language required.
Author Response
Response to the Reviewers’ Comments
Journal: Micromachines
Manuscript ID: micromachines-2424004
Title: A Review of Phase Change Materials and their Potential for Reconfigurable Intelligent Surfaces
Authors: Matos, Randy; Pala, Nezih
We sincerely thank the reviewers for their invaluable feedback. We have revised our manuscript to address the reviewers’ comments. Below are our detailed responses to the comments. The parts added to the revised manuscript are highlighted in yellow.
Reviewer #2:
Reconfigurable intelligent surface has emerged as a promising platform for both wireless RF applications as well as optical ones. This paper reviews the current state of the art of PCMs within the context of RIS, their material properties, their performance metrics, some applications found in literature, and how they can impact the future of RIS. Major revision is necessary according to the following comments.
- The author should compare the basic properties of several main PCMs mentioned in reconfigurable intelligent surfaces, in the form of figure or table.
Response: Thank you for the recommendation. We present the resistivity characteristics for the PCMs in Fig. 1. Additionally, Table 1 was added to present the optical transition characteristics. Table 2 also presents notable applications of PCMs with plenty of references.
- In the introduction section, the author should point out the differences between this review and similar topic reviews, highlighting the importance and timeliness of this review.
Response: Thank you for the recommendation. The introduction was modified to address this point. Please see the highlighted section in page , lines 89-105.
- The last paragraph of the introduction cannot simply describe the basic structure of this review. Major revision is very necessary referring to other reviews.
Response: Thank you for the recommendation. We provided a summary of the recent reviews with proper references and revised the last paragraph of the introduction section. Please see the highlighted section in page 3, lines 89-105.
- A summary should be added at the end of each subsection.
Response: Thank you for the recommendation. A brief summary was added to the end of each subsection and is highlighted.
- Pay attention to the logical coherence between chapters and paragraphs.
Response: Thank you for the recommendation. The paper was reviewed for consistency.
- In section 2.2.1 Chalcogenides, the author is more like a detailed list of five literatures [29-33], and should simplify the basic results and express more of his own opinions.
Response: Thank you for the recommendation. The necessary changes were made and are highlighted in the relevant section.
Round 2
Reviewer 2 Report
Accept in present form